# The evolution of host-symbiont dependence

Roberta M. Fisher[1,2], Lee M. Henry[3], Charlie K. Cornwallis[4], E. Toby Kiers[1,*] & Stuart A. West[2,*]

Organisms across the tree of life form symbiotic partnerships with microbes for metabolism, protection and resources. While some hosts evolve extreme dependence on their symbionts, others maintain facultative associations. Explaining this variation is fundamental to understanding when symbiosis can lead to new higher-level individuals, such as during the evolution of the eukaryotic cell. Here we perform phylogenetic comparative analyses on 106 unique host–bacterial symbioses to test for correlations between symbiont function, transmission mode, genome size and host dependence. We find that both transmission mode and symbiont function are correlated with host dependence, with reductions in host fitness being greatest when nutrient-provisioning, vertically transmitted symbionts are removed. We also find a negative correlation between host dependence and symbiont genome size in vertically, but not horizontally, transmitted symbionts. These results suggest that both function and population structure are important in driving irreversible dependence between hosts and symbionts.

[1] Department of Ecological Science, Faculty of Earth and Life Sciences, Vrije Universiteit, De Boelelaan 1085-1087, 1081 HV Amsterdam, The Netherlands. [2] Department of Zoology, University of Oxford, Oxford OX1 3PS, UK. [3] School of Biological and Chemical Sciences, Queen Mary University of London, London E1 4NS, UK. [4] Department of Biology, Lund University, Sölvegaten 37, 223 62 Lund, Sweden. * These authors jointly supervised this work. Correspondence and requests for materials should be addressed to R.M.F. (email: roberta.fisher@bio.ku.dk).

Symbioses play a fundamental role in ecosystem functioning, organismal health and the evolution of biological complexity. Symbionts allow hosts to live in habitats they would otherwise be excluded from, utilize inaccessible nutrients and capture novel forms of energy[1,2]. While these associations are globally ubiquitous, hosts show huge variation in their dependence on symbiotic partners (Fig. 1). In some cases, hosts and their symbionts become so tightly associated that this has resulted in a major evolutionary transition to a more complex life form, such as the eukaryotic cell. Ultimately, these evolutionary processes depend on the extent to which hosts and symbionts mutually benefit from associations, and how these can be enhanced by the evolution of co-adaptations that lead to greater dependence. Extreme dependence is characterized by physical and genomic integration between host and symbiont[3,4]. For example, cicadas and their nutrient-provisioning endosymbionts function as a metabolic unit and cannot survive without one another[2,5]. In contrast, legumes and $N_2$-fixing rhizobia maintain a facultative relationship, engaging in symbiosis only in certain environments, and benefits to the host can depend strongly on context[6]. Understanding this variation in dependence is key to explaining the evolutionary significance of symbioses, such as why major evolutionary transitions have occurred on certain branches of the tree of life, but not on others[7,8].

It has been hypothesized that both symbiont function and the route by which symbionts are transmitted between hosts (vertically versus horizontally) may be important factors in explaining variation in host dependence across symbioses[9]. Symbiont function is important because it will influence the costs and benefits of engaging in symbiosis. Hosts might be expected to evolve higher dependence on symbionts providing nutrients, such as vitamins[10,11], that are utilized every generation as opposed to benefits only provided in certain environments, such as defence against parasitoids or pathogens[12]. The route via which symbionts are transmitted is also expected to be important because theory predicts it will influence the extent to which symbionts fitness interests are aligned with their host. Vertical transmission of symbionts from a parent to their offspring leads to a shared reproductive fate between symbionts and their hosts that favours cooperation[13–17]. In contrast, horizontal transmission, such as acquiring symbionts from the environment, disrupts this shared fate and can lead to multiple symbiont lineages per host, with competition between symbionts resulting in greater host exploitation. In addition, vertical transmission means hosts consistently inherit the same lineage of symbionts, which allows them to become co-adapted and dependent on their presence[7–9,18].

There have, however, been no quantitative tests as to whether these hypotheses can explain the variation in host dependence across the tree of life. This is problematic because it is not clear that the current data support the theoretical predictions. For example, there are cases in which hosts show high levels of dependence on horizontally transmitted symbionts and low dependence on vertically transmitted symbionts (Fig. 1). The giant marine tubeworm *Riftia pachyptila* obtains its nutritional symbionts horizontally from the environment each generation, but is obligately dependent on them[9,19]. In contrast, aphids transmit their defensive symbiont *Regiella insecticola* vertically with high fidelity, but their removal causes no appreciable fitness cost to the host[20,21]. Are these rare exceptions, or is existing theory unable to explain the variation in host dependence? Another potential problem is that many well-studied examples of symbiosis are based on closely related hosts, such as the many species of facultative symbionts harboured by aphids. This could lead to patterns across species being driven by these commonly studied taxa.

We carried out a phylogenetically based comparative study to determine the correlates of host dependence. We measured host dependence from published experiments in which the effect of symbiont removal on host fitness was measured. We defined symbionts as any species of host-associated bacteria where a beneficial effect had been reported, and thus we did not include reproductive manipulators such as some *Wolbachia* species or host-bound obligate pathogens. We used the change in host fitness as a quantitative measure of the extent to which hosts were dependent on their symbionts, with larger decreases in fitness representing greater dependence.

In total, we found data on 106 symbioses, formed between 58 bacterial symbionts and 89 host species, including insects, plants, fungi, molluscs, arachnids and worms (Fig. 2). Some symbiont species are found in multiple hosts, and some hosts have multiple symbionts, resulting in partially overlapping symbionts and hosts in our data set. These studies examined the fitness consequences of symbiont removal in a number of different ways, such as longevity, fecundity and host size, and so we also analysed whether there was

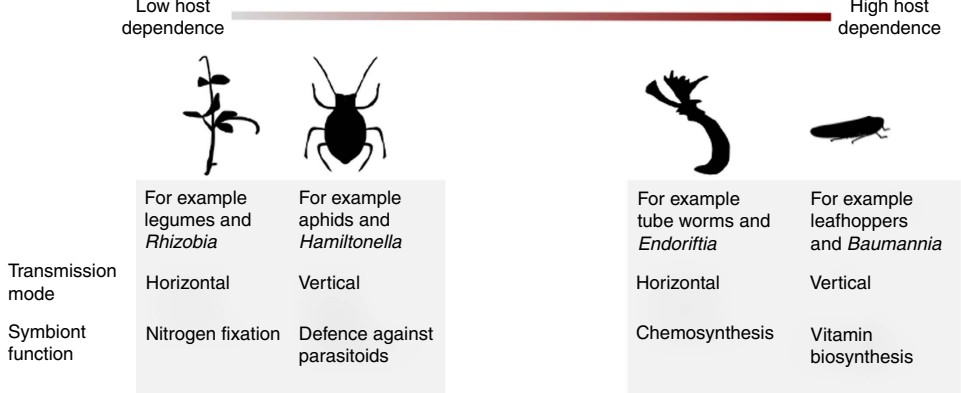

**Figure 1 | Variation in host dependence on symbionts and their predominant mode of transmission.** The degree to which hosts are dependent on bacterial symbionts for survival and reproduction varies hugely across taxa. Some hosts and symbionts have facultative relationships, where neither partner is dependent on each other. This may be because hosts only need symbionts in certain conditions, for example, legumes and *Rhizobia,* or because symbionts are only present in some of the host population, for example, aphids and *Hamiltonella*. In contrast, some hosts are entirely dependent on symbionts for nutrition either because they live in extreme environments, for example, deep-sea tube worms and *Endoriftia*, or because they have restricted diets, for example, leafhoppers and *Baumannia*. Image of *Graphocephala coccinea* sourced from phylopic.org, courtesy of Melissa Broussard available under a Creative Commons license.

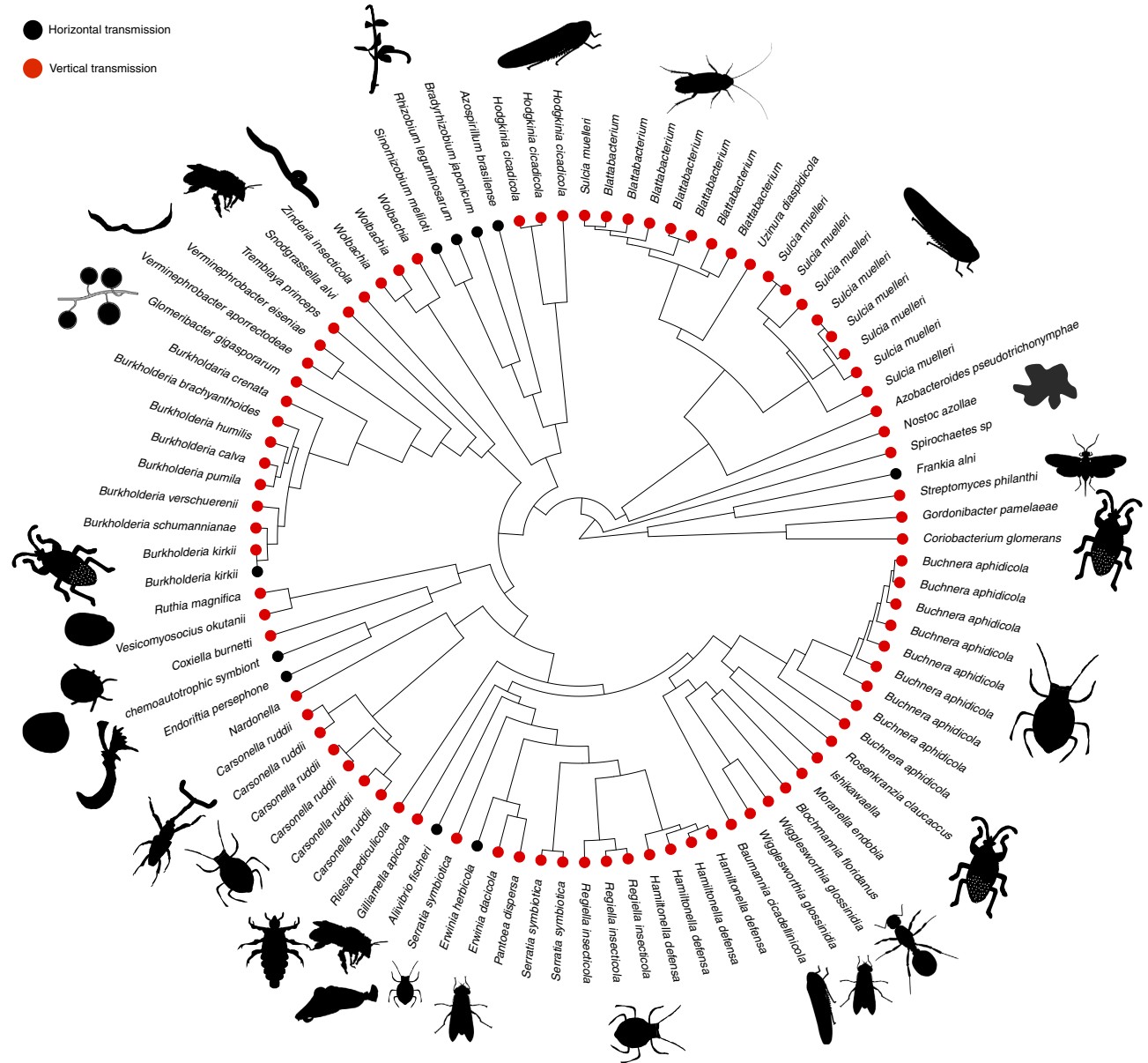

**Figure 2 | Bacterial symbiosis across the tree of life.** A phylogeny of bacterial symbionts in our database built using an ∼1,500 bp region of the 16S rRNA gene, with images of their host groups. Vertically transmitted symbionts are shown with red circles and horizontally transmitted symbionts are shown with black circles. Multiple species names at the tips are distinct lineages found in different host groups. Images of hosts are not intended to be exact representations of the host species, but more of the general host group (for example, wasp, worm and plant). Where there is no image, this is because a suitable representation of the host species was not available. Host images sourced from Phylopic (phylopic.org): *Graphocephala coccinea* courtesy of Melissa Broussard; *Hippoboscoidea* fly courtesy of Gareth Monger; *Sepioidea* squid courtesy of David Sim and T. Michael Keesey; *Uropodina* tick courtesy of Birgit Lang; and *Brugia* nematode courtesy of Gareth Monger; all available under a Creative Commons license.

any correlation between host fitness drop and type of fitness measure. We collected data on symbiont function, whether the symbiont is intra- or extracellular, the age of the symbiosis, symbiont genome size and transmission mode. For transmission mode, we estimated the average rate of vertical/horizontal transmission per generation, because that is what is predicted to be important for determining host–symbiont cooperation levels[12,13,15]. We therefore defined symbionts that are predominantly transmitted from parent to offspring as vertically transmitted, even when there are rare events of horizontal or biparental transmission between hosts (see Methods).

We found that a higher host dependency, measured as a greater drop in host fitness when symbionts were removed, was associated with vertical transmission, nutritional functions and small genome size. These results suggest that specific factors may drive host dependence to evolve in predictable ways across diverse symbioses.

## Results

**Symbiont transmission.** We first tested whether transmission mode was correlated with the level of host dependence. We found that the removal of vertically transmitted symbionts resulted in a fitness reduction twice as large as that with horizontally transmitted symbionts (52% versus 21%; Fig. 3a and Supplementary Table 1). This was not an artefact of host or symbiont evolutionary

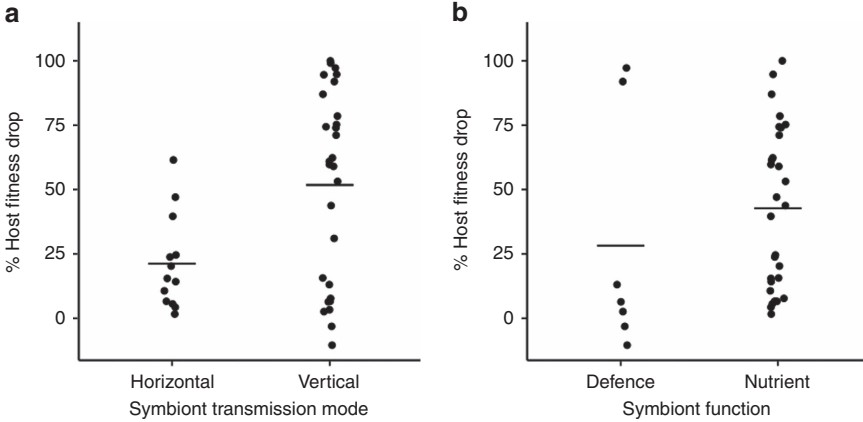

**Figure 3 | Transmission mode and symbiont function correlate with host dependence.** Each data point represents the mean % reduction in host fitness when its symbiont is removed from each unique symbiosis. The horizontal lines represent the mean % host fitness reduction. (**a**) Hosts tend to be more dependent on vertically transmitted symbionts. Plotted $N_{symbioses} = 38$; (**b**) Hosts tend to be more dependent on nutritional compared to defensive symbionts. Plotted nutritional, $N_{symbioses} = 28$, and defensive, $N_{symbioses} = 7$.

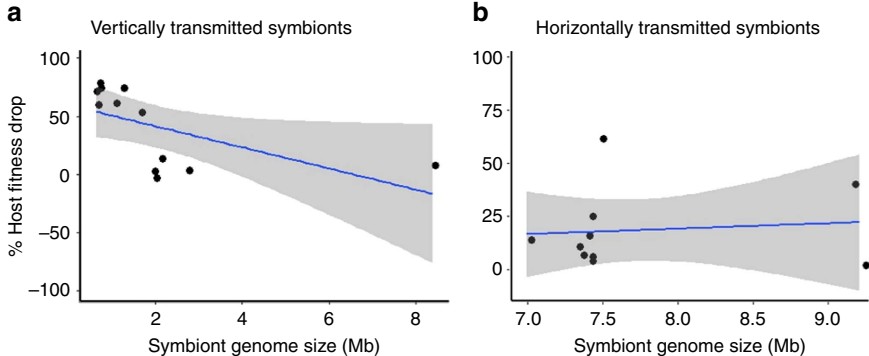

**Figure 4 | Correlated evolution of host dependence and symbiont genome size.** Each data point represents the mean % host fitness reduction for a unique host-symbiont combination with 95% confidence intervals. (**a**) Vertical transmission: BPMM; posterior mode for slope = − 0.14; CI = − 30.54 to 6.65; $N_{symbioses} = 12$. (**b**) Horizontal transmission: BPMM; posterior mode for slope = 1.03; CI = − 31.51 to 24.50; $N_{symbioses} = 10$.

history[22,23], as the difference between horizontally and vertically transmitted symbionts remained after accounting for both host and symbiont phylogenetic relationships (symbiont phylogeny: Bayesian phylogenetic mixed model (BPMM), posterior mode ($\beta$) = − 36.58, 95% credible interval (CI) = − 67.54 to 8.45, pMCMC = 0.01; host phylogeny: BPMM, $\beta$ = − 39.33, CI = − 67.96 to 9.13, pMCMC = 0.003; Supplementary Tables 2 and 3). Together, these analyses support the prediction that vertical transmission of symbionts selects for greater host dependence.

**Symbiont function.** We divided symbionts into those that provide nutritional benefits to their hosts, such as production of amino acids, vitamins or photosynthates, and those that provide a defensive function, such as protection against pathogens or parasitoids. We found that the removal of symbionts that provide nutritional benefits to their hosts was correlated with greater host fitness reductions than removal of defensive symbionts (43% versus 28% drop; Fig. 3b). This result held when accounting for host phylogenetic relationships (BPMM, $\beta$ = − 24.93, CI = − 63.44 to 4.84, pMCMC = 0.01; Supplementary Table 3) and phylogenetic relationships between symbiont species (BPMM, $\beta$ = − 28.75, CI = − 66.80 to 2.79, pMCMC = 0.03; Supplementary Table 2). The percentage of variation in symbiont function explained by symbiont phylogeny was 75% (phylogenetic heritability estimated using BPMM; CI = 63 to 86) and by host phylogeny was 69% (BPMM; CI = 44 to 81), suggesting that

closely related symbiont species tend to perform similar functions for their hosts, and that closely related hosts have similar needs.

We then tested whether symbiont function was correlated with host dependence within each transmission mode. Within vertically transmitted symbionts, we found that hosts suffered three times the reduction in fitness when symbionts that provide nutritional benefits were removed as opposed to those involved in host defence (61% host fitness reduction when nutritional symbionts removed). This is compared to a 19% fitness reduction when horizontally transmitted nutritional symbionts were removed. This suggests that the function of symbionts and host dependence may be more correlated when symbionts are vertically transmitted compared to when they are horizontally transmitted. Overall, our results point to hosts being most dependent on vertically transmitted, nutritional symbionts. This appears to be a relatively general pattern as the diversity of host species tested included fungi, termites, tsetse flies and aphids, encompassing symbionts that provide a wide variety of nutritional functions.

**Host dependence and genome evolution.** When the reproductive fate of hosts and symbionts become aligned through vertical transmission, the potential for co-adaptation is increased. One predicted consequence of this is that certain symbiont genes may become redundant because they are no longer needed in the host environment and therefore will be lost through drift

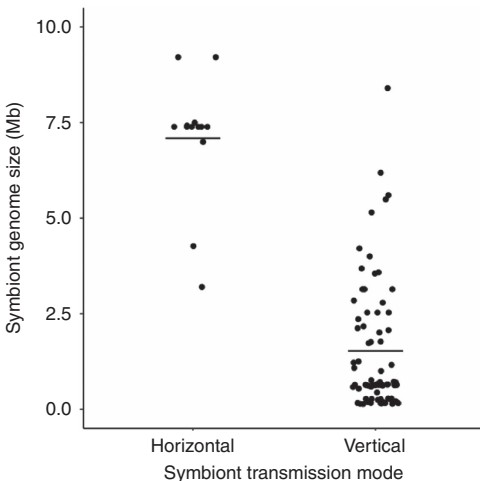

**Figure 5 | Vertically transmitted symbionts have smaller genomes.** Each data point represents the genome size for the symbiont in a unique host–symbiont combination. The horizontal bars represent mean genome sizes for each transmission mode. Horizontal: $N_{symbioses} = 13$ and $N_{datapoints} = 125$; vertical: $N_{symbioses} = 55$ and $N_{datapoints} = 253$.

and selection[4,18,24]. For example, the leafhopper *Macrosteles quadrilineatus* houses the endosymbiont *Nasuia deltocephalinicola*, which has the smallest observed bacterial genome[25]. The expectation is that highly dependent host–symbiont relationships are characterized by small symbiont genomes[26], but tests of this hypothesis have not been quantitative, nor tested across divergent hosts. We therefore determined whether host dependence was correlated with symbiont genome size.

We found that the genome sizes of vertically transmitted symbionts were significantly smaller than those of horizontally transmitted symbionts, highlighting the role of transmission route in genome evolution (Fig. 4; symbiont phylogeny: BPMM, $\beta = -5.56$, CI $= -6.74$ to $-4.10$, pMCMC $= <0.001$; host phylogeny: $\beta = 4.21$, CI $= 3.09$ to 5.56, pMCMC $= <0.001$; Supplementary Tables 2 and 3). We also found a negative correlation between symbiont genome size and host fitness reduction within vertically transmitted symbionts (symbiont phylogeny: BPMM, $\beta = 1.03$, CI $= -31.51$ to 61.49; host phylogeny: $\beta = -0.11$, CI $= -23.30$ to 8.47), but not in horizontally transmitted symbionts (symbiont phylogeny: BPMM, $\beta = -0.14$, CI $= -30.54$ to 6.65; host phylogeny: $\beta = 0.15$, CI $= -38.88$ to 49.94), although the difference between these slopes was not significantly different (Fig. 5a,b; symbiont phylogeny: BPMM, $\beta = 1.17$, CI $= -0.98$ to 54.85, pMCMC $= 0.25$; host phylogeny: BPMM, $\beta = 0.72$, CI $= -42.59$ to 58.77, pMCMC $= 0.37$; Supplementary Tables 4 and 5). It has previously been shown that symbionts have reduced genomes compared with free-living bacteria[26–28], but the link with transmission mode had not been formally tested. These results are consistent with the hypothesis that vertical transmission aligns reproductive interests in a way that favours the correlated evolution of genome reduction and host dependence.

The correlation between dependence and genome size could, alternatively, be explained by time. The longer a symbiont and host have been associated the more we would expect the evolution of dependence and genome degradation to occur through genetic drift and selection, particularly in vertically transmitted symbionts. Estimates of the time of origin of symbioses are difficult to obtain. However, we gained data on 54 symbioses from our full data set and found that among these examples, there was no significant association between age of symbiosis and host dependence (symbiont phylogeny: BPMM,

$\beta = 0.14$, CI $= -0.27$ to 0.47, pMCMC $= 0.30$; host phylogeny: BPMM, $\beta = -0.0007$, CI $= -0.50$ to 0.36, pMCMC $= 0.43$; Supplementary Tables 6 and 7).

**Intracellular and obligate symbionts.** Many bacterial symbionts are housed intracellularly in host cells[18], and it is possible that this could play an important role in driving host integration and dependence. For example, intracellular symbionts are less exposed to life outside the host environment, and this could select for more cooperative symbionts and more dependent hosts. Therefore, we included whether symbionts were intracellular or extracellular as an explanatory variable in all of our analyses. We found that hosts were not significantly more dependent on intracellular symbionts compared to extracellular ones (symbiont phylogeny: BPMM, $\beta = 19.10$, CI $= -3.88$ to 49.16, pMCMC $= 0.07$; host phylogeny: BPMM, $\beta = 20.16$, CI $= -6.31$ to 44.57, pMCMC $= 0.06$; Supplementary Tables 2 and 3). Furthermore, we found that intracellularity and vertical transmission are not significantly correlated over evolutionary time (phylogenetic correlation $= 0.009$, CI $= -0.05$ to 0.08, pMCMC $= 0.43$), suggesting that transmission route can evolve relatively independently from whether symbionts are extracellular or intracellular.

We also tested whether host dependence was correlated with the dependence of the symbiont on the host, which can be obligate (only found in hosts) or facultative (found both in host and free-living state). We found that hosts were more dependent on obligate symbionts (Supplementary Fig. 1; symbiont phylogeny: BPMM, $\beta = -23.95$, CI $= -48.36$ to 10.64, pMCMC $= 0.02$; host phylogeny: BPMM, $\beta = 61.22$, CI $= -0.79$ to 136.60, pMCMC $= 0.006$), suggesting that the evolution of host and symbiont dependence are linked. However, we did not include this in analyses with transmission mode, as (by definition) all horizontally transmitted symbionts are also facultative, as they retain the ability to survive outside the host.

**Robustness of our results.** Our study includes data from experiments that measured the fitness of hosts, with and without symbionts, in different ways. Usually, host fitness was measured either as the size of hosts or offspring, the survival of offspring to adulthood, developmental time, resistance to pathogens or as fecundity. To determine whether these measurements influenced our findings, we tested the role of different experimental approaches and found no differences in host fitness drop among methods for measuring host fitness (Supplementary Tables 8 and 9, and Supplementary Fig. 2), and no significant interaction between the way in which host fitness was measured and transmission mode (excluding developmental time, survival and resistance due to lack of replication: BPMM, $\beta = -21.01$, CI $= -38.68$ to 7.68, pMCMC $= 0.09$, $N_{symbiont\ species} = 21$). We also ran analyses on a restricted data set using only measures of host fitness based on fecundity, as we expect this to be the most reliable and relevant measure of fitness. Using this data set, the difference in host fitness drop between vertically and horizontally transmitted symbionts remained the same as when using the complete data set, again supporting the robustness of our results (symbiont phylogeny: BPMM, $\beta = -50.89$, CI $= -85.68$ to 4.41, pMCMC $= 0.02$; host phylogeny: BPMM, $\beta = -43.32$, CI $= -86.98$ to $-12.12$, pMCMC $= 0.0006$; Supplementary Tables 10 and 11).

**Discussion**
We used the drop in host fitness when symbionts were removed as a measure of host dependency. We found a higher host dependency on vertically transmitted and nutritional symbionts

(Fig. 3). In addition, we found that the genome size of vertically transmitted symbionts was smaller than that of horizontally transmitted symbionts (Fig. 5) and that host dependency was negatively correlated with symbiont genome size in vertically transmitted symbionts (Fig. 4).

Although we examined a diverse range of hosts and symbionts (Fig. 2), there are several limitations with the available data. First, host dependence data are reliant on removal of symbionts from hosts, which is not always possible as they are often housed within host bacteriocytes where antibiotic treatment is ineffective or because of antibiotic toxicity[29]. However, such combinations are likely to show high levels of dependence and be vertically transmitted, and so this limitation probably makes our results more conservative.

Second, there is an emphasis in the literature on intracellular, vertically transmitted symbionts. This may be due to sampling bias, for example, marine systems are under-represented and insect systems are over-represented, but it may also be a reflection of the rarity of stable horizontal symbioses in terrestrial systems. What matters for comparative analyses is the number of phylogenetically independent contrasts between different transmission modes, and so our analyses are controlling for, not giving undue weight to, where we have multiple closely related species where transmission mode does not vary.

Another challenge is to accurately estimate fitness consequences of losing symbionts. To identify general patterns of host dependence across species, we included as much data as possible. However, this meant including experiments that differ in their methods and the environment in which they measure host fitness, potentially introducing extra variables responsible for the variation in host dependence. This problem is particularly challenging for defensive symbionts, where tests must be carried out with 'natural' predator or pathogen pressure, which can be hard to ascertain. These issues emphasize that caution is required when interpreting the influence of symbiont function (Fig. 3b), as it may be an artefact of how researchers have estimated the consequences of removing defensive symbionts. For example, if the natural enemy is introduced at an unnatural density, or in unusual environmental conditions, then misleading fitness consequences will be obtained. Our data are unable to address whether host dependence is context-dependent, as experiments often only measure host fitness in one environment. However, there is no expectation that any context dependence would be specific to either vertically or horizontally transmitted symbionts, and so this would be unlikely to affect our main conclusions.

Another factor is that the strength of the relationships we found may be contingent on the environment in which the symbiosis evolved, for example, in marine versus terrestrial environments. In marine systems, horizontal acquisition of symbionts from surrounding sea water is common, as seen in *Bathymodiolus* mussels, but hosts may still rely on symbionts for fundamental needs, such as nutrition[19,30].

Our results support the hypothesis that vertical transmission plays a decisive role in driving the evolution of dependence across divergent host clades. Transmission mode matters because it can provide a powerful mechanism for aligning the reproductive interests of partners[13,15–17]. Furthermore, the broad comparative patterns that we have found across symbiotic partnerships are consistent with experimental studies showing that spatial structuring or reduced dispersal can favour increased cooperation within mutualisms[31,32]. Comparative and experimental studies allow for different but complementary issues to be examined[33]. Our comparative analyses allow us to test for broad-scale patterns across species, but they do not demonstrate causation. In contrast, experimental studies can determine causation, but only in single cases. Taken together, comparative and experimental data provide strong support for a role of transmission in driving the evolution of host dependence.

More generally, it has been argued that vertical transmission has played a key role in explaining when hosts and their symbiotic partners may evolve into a new, single integrated organism[8]. Such major evolutionary transitions between species can occur, but they are rare and require strict conditions[3,7,34–36]. The evolution of eukaryotic cells can be conceptualized as an extreme example. In cases of both mitochondria and chloroplasts, once free-living proteobacterial and cyanobacterial ancestors became integrated within cells and began to be transmitted only vertically[37,38]. Our data also show that function plays a role in host dependence, with hosts evolving higher dependence on symbionts providing nutrients that are utilized every generation, as opposed to defensive benefits that are likely to only be needed in certain environments. Our results suggest that host–symbiont dependence may evolve in predictable ways across the tree of life, allowing us to better understand which partnerships have led to major evolutionary transitions[3,8].

## Methods

**Literature search.** We compiled a database on host–bacterial symbioses, including data on transmission mode, symbiont function, whether the symbiont was intra- or extracellular, symbiont genome size, whether symbionts were obligate or facultative, and measures of host fitness with and without symbionts. We found papers by (1) searching combinations of the following key words using Papers 2 (covering Scopus, Web of Knowledge, JSTOR, Google Scholar and PubMed)— bacteria, symbiont, symbiosis, mutualism, fitness, removal, elimination, fecundity, dependence, obligate, facultative and aposymbiosis; (2) searching several key reviews[9,18,24,28]; and (3) forward and backward searches from resulting papers.

Our data set contains published analyses that measured host fitness with and without their bacterial symbionts from experimental manipulations, found through searching the literature as described in the main methods. We eliminated papers that (1) did not report fitness measurements with and without bacterial symbionts, (2) where the PDF was unavailable or not available in English and (3) studies involving the introduction of a novel symbiont to a naive host. In cases where the required data were not included in the paper, we contacted the corresponding author to try and obtain data to include in our analyses. Overall, this literature search returned over 200 papers, and resulted in a database including a total of 378 fitness change measures, from 84 studies.

**Data collection.** We used a classification scheme to divide symbionts in a manner that theory predicts will be important. Evolutionary theory predicts that selection for cooperation will depend on the relative rate of horizontal and vertical transmission per generation, rather than over evolutionary time[13,14]. Many authors have argued that transmission mode is a continuum ranging from strict vertical to purely horizontal transmission. Here we simplify this spectrum into two broad categories for several reasons. From a theoretical perspective, it is transmission over an ecological timescale that matters for the evolution of symbiont cooperation[7–9] and so we classified a species as vertically transmitted even in cases where there are rare events of horizontal transmission of symbionts either between hosts (for example, *Hamiltonella*) or through rare biparental transmission (for example, *Regiella*[39,40]). We classified a symbiont as horizontally transmitted when hosts predominantly acquire symbionts from the environment or conspecifics, rather than parents[9]. Second, the ideal way to classify transmission mode would be to use quantitative data on degree of fidelity, but these data are difficult to measure and very rare in the literature. Therefore, we have classified species as having vertical transmission if they typically transmitted symbionts from parent to offspring, and as horizontally transmitted when hosts acquire symbionts from the environment or unrelated conspecifics, rather than from a parent[4]. When the same symbiont was found in multiple hosts, but details about transmission could only be found for one host, we assumed that the transmission mode was the same in all hosts.

We classified each symbiont as having either broadly defensive, nutritional or an unknown function. Nutritional functions ranged from production of amino acids or vitamins to photosynthesis and nitrogen fixation, and defensive functions ranged from protection against pathogens to antibiotic production. We defined symbionts as obligate if they are always host-associated and never found in a free-living state, whereas we defined facultative symbionts as being able to survive outside of their host and therefore sometimes found free-living. We then further classified each symbiont as being intracellular (present within host cells) or extracellular (present outside of host cells, for example, in the gut), to allow us to control for this in our analyses.

We gathered all symbiont genome size data in GenBank. If we found multiple entries for a symbiont species, we included the entry that corresponded to the host species in our database. If we found multiple entries with different genome sizes we

took the mean for that particular symbiont (Supplementary Table 14). If the genome for a symbiont species could not be found on GenBank, we searched the literature using Papers 2 (as described above) and used the most recently published complete genome.

We calculated the percentage change in host fitness as follows: host fitness with symbiont − host fitness without symbiont/host fitness with symbiont × 100. This allowed us to standardize changes in host fitness across many studies where this was measured in a variety of different ways, including survival to adulthood, fecundity and size or mass. For several studies we calculated the overall mean ourselves, from raw data or graphical plots (Supplementary Table 13). We categorized the way in which host fitness was measured in each study as either (1) fecundity (that is, number of offspring), (2) size/mass of hosts and/or offspring, (3) developmental time of offspring, (4) survival of offspring or (5) resistance against pathogens or parasites.

We searched for data on the age of symbiosis by using published time-dated phylogenies where the origin of association between host and symbiont had been determined. If this was not possible, we used estimates of the age of symbiosis determined from the fossil record.

### Statistical methods.

We used Bayesian general linear models (BPMMs) with Markov Chain Monte Carlo (MCMC) estimation with host fitness reduction and symbiont genome size both fitted as Gaussian response variables. We fitted transmission route (two-level factor), function (three-level factor), whether symbionts were intra- or extracellular (two-level factor) and whether symbionts were obligate or facultative (two-level factors) as fixed effects. To assess the robustness of our results, we ran an additional analysis where we included the way in which host fitness was measured as a fixed effect (four-level factor: fecundity, size/mass, survival and developmental time) and host fitness reduction as the response variable. We weighted each data point by the inverse of the sample size in the experiment where the data were obtained from (MCMCglmm code: mev = 1/N). We removed the global intercept to allow for host fitness drop and genome size to be estimated separately and fitted $2 \times 2$ unstructured phylogenetic and residual covariance matrices as random effects to estimate correlations between host fitness drop and symbiont genome size.

We accounted for shared ancestry between hosts by fitting a variance–covariance matrix of phylogenetic distances between host species generated from the host phylogeny as a random effect. To account for shared ancestry between symbionts, we used the same approach but used a phylogeny of symbiont species. We included random effects to control for multiple measures of fitness per species (symbionts or hosts) and multiple measures per study.

To test whether the relationship between host fitness reduction and symbiont genome size was different for each transmission mode, we ran a model with transmission-specific correlations. We did this using a multi-response BPMM, where separate phylogenetic and residual variance–covariance matrices for host fitness reduction and symbiont genome size were fitted for vertically and horizontally transmitted symbionts using the at.level coding in MCMCglmm (see Supplementary Data 1 for details of the R code). From these models we calculated the correlations between host fitness reduction and symbiont genome size (Cov(% fitness drop, genome size)/sqrt(Var(% fitness drop) × Var(genome size)) for vertically and horizontally transmitted symbionts. We tested whether the correlation was significantly different between transmission modes by examining whether the 95% CI of the difference between the correlations spanned 0, and calculating the percentage of iterations, where the correlation for vertically transmitted symbionts was greater than that for horizontally transmitted.

We ran each analysis for 5,500,000 iterations with a burn-in of 500,000 and thinning interval of 5,000 to minimize any auto-correlation between posterior samples. We checked that models converged by visually observing trace plots of MCMC chains, using Gelman–Rubin tests (R package 'coda')[41,42] and confirming Geweke diagnostics were <2. The prior settings used for each analysis are specified in the Supplementary R code. For all random effects we used inverse-Wishart priors ($V = 1$, nu = 0.002), which led to well-mixed chains with low auto-correlation.

The parameter estimates we report in the results section and in results tables are the posterior mode and 95% CIs (lower CI–upper CI). We calculated estimates of the differences between the levels of fixed effects from a posterior distribution created by subtracting the estimates for each level obtained during each MCMC iteration (labelled 'difference' in results tables). We considered parameter estimates statistically significant when 95% CIs did not include 0 and pMCMC values (number of simulated cases that are >0 or <0 corrected for finite number of MCMC samples) were <0.05.

### Building and pruning phylogenies.

We estimated the phylogenetic relationship of the 61 bacterial symbionts in the full data set using an ∼1,500 bp region of the bacterial 16S rRNA gene, which were downloaded for the SILVA RNA database. We aligned sequences with MUSCLE and edited in the alignment software Geneious 8.1.8 (ref. 43). We determined the best-fitting model of evolution using MEGA5's model selection algorithm[44] and generated a maximum likelihood phylogeny for the bacterial lineages using the on-line PhyML server[45]. We bootstrapped the symbiont phylogeny 100 times and rooted to Thermus thermophilus, which is basal to all the bacterial lineages presented in this study.

We built the host phylogeny using the R package 'rotl' (https://cran.r-project.org/web/packages/rotl/index.html) which combines taxonomic information and published phylogenies from the Open Tree of Life[46] (http://www.opentreeoflife.org). This creates a synthetic tree, by matching species names in our database to taxon names in the Open Tree of Life and uses the function 'tol_induced_subtree' to retrieve phylogenetic relationships and produce a 'phylo' object. We then pruned both phylogenies using drop.tip() so that they only contained the host or symbiont species needed for each analysis.

### Sensitivity analyses.

To estimate the sensitivity of our results to shared ancestry between both host species and symbiont species, we ran analyses where the host phylogeny was used to fit a phylogenetic variance–covariance matrix as a random effect and a second where symbiont phylogeny was used. In both sets of analyses phylogenetic and residual correlations were calculated as covariance between traits xy/square root (variance in trait x × variance in trait y). We also estimated the amount of variation in host fitness drop explained by shared ancestry between host species and between symbiont species (phylogenetic heritability) as phylogenetic variance/(residual + phylogenetic variance) × 100. To test how robust our results were to different measures of fitness, we also ran an analyses on a restricted data set where only fitness measures based on fecundity were used ($N_{data points} = 102$).

We also ran analyses to check the effect of the way in which host fitness was measured on fitness drop. This allowed us to check that how host fitness was measured did not affect our results. First, we ran the multi-response analysis described above on a restricted data set where host fitness was only measured as fecundity (Supplementary Tables 10 and 11). Second, we ran an analysis with the way in which host fitness was measured (four levels, excluding resistance due to lack of replication) as a fixed effect and host fitness drop as a response (Supplementary Tables 8 and 9).

Finally, to ensure our results were not affected by potential examples of biparental transmission of symbionts, we ran our main analysis again, reclassifying Verminephrobacter and R. insecticola as horizontal. Both these species are potentially biparentally transmitted[40,47], and so their transmission has a horizontal component. However, when we reassigned these species as horizontally transmitted, our result stayed the same (Supplementary Table 12). We found that vertically transmitted symbionts resulted in larger host fitness reductions when they were removed compared to horizontally transmitted symbionts (BPMM, $\beta = -26.64$, CI $= -53.44$ to $-6.61$, pMCMC = 0.002).

**Data availability.** The data used in this study are available from the corresponding author on request.

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

## Acknowledgements

We thank Angela Douglas, Anna Himler, Martha Hunter, Takahiro Hosokawa, Ned Ruby, Hassan Salem and Christoph Vorburger for data; Philip Downing, Anouk van't Padje and Shana Caro for comments on the manuscript. This research was supported by Netherlands Organisation for Scientific Research Vidi and Aspasia Grant 864.10.005 (to E.T.K.) and European Research Council ERC Grant Agreement 335542 (to E.T.K.); the Swedish Research Council and Knut and Alice Wallenberg Foundation (to C.K.C.); European Research Council (to S.A.W.); Natural Environment Research Council (to R.M.F.); and Natural Environment Research Council IRF NE/M018016/1 and Marie Curie IEF 626616 (to L.M.H.).

## Author contributions

R.M.F., E.T.K. and S.A.W. designed the study; R.M.F. collected the data; R.M.F., L.M.H. and C.K.C. carried out the analyses; all authors wrote and commented on the manuscript.

## Additional information

**Competing interests:** The authors declare no competing financial interests.

