## [Peer Review File · Nature Communications]

Reviewers' comments:

Reviewer #1 (Remarks to the Author):

The authors try to answer the question: "Which factors drive variation in dependence of hosts on symbionts across different symbioses?" However, the so-called causal factors in reality are a mixture of causes and consequences of dependence. Take the relationship between genome size and dependence. On the one hand, it seems likely that a reduction in symbiont genome size only can occur after dependence has evolved, since natural selection during independent stages of the bacteria will favour the full genome size. But on the other hand, after some dependence has evolved, subsequent loss of symbiont genes will make dependence bigger. So even though a correlation between genome size and dependence seems likely, as the authors confirm for vertically transmitted symbionts, such a correlation does not provide an answer to the question addressed.

Despite this problem, the whole text is written as if a phylogenetic comparative analysis will enable the discovery of causal factors for differences in dependence of hosts on symbionts across different symbioses. At best, such an analysis will provide correlations between dependence and certain factors. I find it misleading to present those correlations as causal relationships. As an example, transmission mode and symbiont function have the strongest correlation with symbiont dependence and this finding is described (in the abstract) as: "We found that both transmission mode and symbiont function were important in explaining (my italics) *the evolution of host dependency*, with the reductions in host fitness being greatest when nutrient-provisioning, vertically transmitted symbionts were removed." I think this interpretation does not do justice to alternative causal explanations. For example, it seems likely that certain nutritional dependences have evolved after vertical transmission evolved, securing long-term interaction, and facilitating loss of redundancy in functions between both partners. In fact, flipping cause and consequence could make perfect sense ("We found that host dependency was important in explaining the evolution of both transmission mode and symbiont function, with the reductions in host fitness being greatest when nutrient-provisioning, vertically transmitted symbionts were removed.")

Detailed points:

1. It should be pointed out that vertical transmission must be uniparental in order to be vertical (and not just "from parent to offspring"), otherwise there is a horizontal component.
2. Within the category of beneficial interaction, the authors focus on dependence. Why host-symbiont dependence? The interesting trait would be 'beneficial' or 'detrimental'. But this is hard to measure.
3. The whole story is based on a phylogenetic comparative analysis of 110 host-bacterial symbioses. This sounds like a lot, but the strength of such analysis depends on the quality of the data. I think the authors need to point out the uncertainty with the phylogenetic estimate, and how robust their analysis is. Furthermore, how well has dependence been studied for the different examples included in this study.
4. Dependence or dependency? Both terms are used in the manuscript.

Reviewer #2 (Remarks to the Author):

The manuscript explores evolutionary correlates of dependency in bacterial symbioses, a topic of broad significance in biology. The authors present several very interesting findings, including links between transmission mode and the evolution of dependency and genome size reduction. Overall, I found this study to be very compelling, nicely executed, well written, and very big picture. I think that it will be a highly cited contribution. However, that being said, I have several major concerns:

Major comments:

In many cases, the number of datapoints used in the analyses is much larger than the number of species included. It was difficult to understand whether symbiont-host pair, species, or 'datapoint' was used as the unit of replication in the models. This made me worry about pseudoreplication and power inflation, as well as skewing results to be more heavily weighted by species that are represented by more papers in the literature. For example, for the analysis with genome size as the response variable and transmission mode as the predictor, 13 horizontally transmitted and 54 vertically transmitted symbioses were included, but 125 and 253 datapoints were used in the analysis, respectively (Figure 5 figure legend). It appears that the species means were not used in the analyses ($n=66$, as presented in the figure), or estimated as part of the model, but that all of the datapoints were used in the analyses ($n=378$) resulting in multiple datapoints per species being treated as independent. Tables S2 and S3 suggest the latter. This feels to me like an over inflation of power. Do results of the paper (in particular, the analyses presented in tables S2 and S3) hold if species are treated as the unit of replication (meaning there would be a maximum of 61 symbiont datapoints and 92 host datapoints in analyses)? Perhaps I am missing something here (and I apologize if this is the case), but this feels to me like a major issue to be resolved.

As the authors mention, the cost/benefit that a host accrues from a given symbiosis changes across different environments. As such, we would expect the 'change in host fitness with symbiont removed' to be a function of the context in which each study was conducted. In particular, it is important to know whether removal experiments occurred in the presence or absence of stressors (i.e., pathogens or parasites in the case of defensive mutualisms, or nutrient stress in the case of nutritional mutualisms). To be fair, the authors touch on this issue nicely at the end of the manuscript for defensive symbionts (but strangely, not nutritional symbionts, stating earlier that the benefit is ubiquitous as a rule, which I disagree with, e.g., nitrogen fixers in high nitrogen environments). However, the authors do not report how their dataset is likely affected by from this issue. What number of studies in this dataset reported the presence of stressors? Is it possible to run analyses on a subset of data using only studies conducted in the face of stressors? Does this change the results?

The methods state (lines 292-294): "We considered parameter estimates statistically significant when 95% credible intervals did not include 0 and pMCMC values (number of simulated cases that are > 0 or < 0 corrected for finite number of MCMC samples) were less than 0.05." However, in several cases, reports are treated as significant when 95% CI

appears to overlaps 0:

(1) For the difference in genome size –dependency relationship slope between vertically and horizontally transmitted symbionts (figure 4A, line 153), the confidence interval overlaps 0.

(2) Again line 111 for the difference in fitness reduction of removing symbionts in vertically vs horizontally transmitted symbionts.

(3) Again line 184

Minor Comments:

Love figures 1 & 2! Very nice looking and clear.

Brief information on how the literature search was conducted is needed in the main methods. Line 262 suggests that the authors used Papers 2 (“we searched the literature using Papers 2 (as described above)”, but this information is only in the supplemental methods.

In figure 4 there is one vertically transmitted symbiont that is an outlier for genome size (I think it is *Rhodococcus rhodnii*, based on the supplementary table). However, this datapoint appears to be missing from figure 5. Why?

Line 121 of supplement states “(see supplementary R code for details)” (also see line 135) but no R code is included. Including the code would really help the readers understand exactly what was done, as well as the replicate results.

If the specific and jargony term ‘evolutionary rabbit holes’ is used in the abstract, it should probably be discussed (or at least defined) in the main text as well.

Reviewer #3 (Remarks to the Author):

This is an interesting paper. Its strength comes from its breadth. I am not aware of any other work which tries to broadly tie so many important factors (transmission mode, symbiosis age, intra- vs. extra-cellularity, etc.) together with such a broad sampling of taxa across the tree of life. The methods all seem reasonable to me, and the conclusions seem solid. Some of the findings are not very surprising, but the authors are correct in saying that while many of these ideas have been out there for a long time, they have never been carefully tested. So there is utility here, and I think this paper will be cited because of that.

I have little to criticize about the paper, aside from two issues. The first relates to the analysis of genome size and host dependence. There is some discussion of the age of the symbiosis being important (and potentially confounding). This is mostly on lines 163-170. The authors conclude that “there was no association between age of symbiosis and host dependency.” I would be very interested to know if this conclusion seems to be affected by symbiont replacement. In general, as the authors find, symbionts which are required by hosts tend to have smaller genomes. But these can sometimes be replaced (see for example the “rabbit hole” paper the authors mention in the abstract). The authors have not discussed symbiont replacement in the paper, but it is known for some of the symbioses

under study, so it might be worth checking if that affects these conclusions.

The second relates to the abundance of insect symbioses in the study. I know the authors corrected for phylogeny in some cases, but should the large number of insect systems be cause for concern here? This might be appropriate, since most animals are in fact insects, but I wonder if randomly down-sampling insect examples might change the results.

Specific point-by-point responses to reviewers comments

Reviewer #1 (Remarks to the Author):

The authors try to answer the question: “Which factors drive variation in dependence of hosts on symbionts across different symbioses?” However, the so-called causal factors in reality are a mixture of causes and consequences of dependence. Take the relationship between genome size and dependence. On the one hand, it seems likely that a reduction in symbiont genome size only can occur after dependence has evolved, since natural selection during independent stages of the bacteria will favour the full genome size. But on the other hand, after some dependence has evolved, subsequent loss of symbiont genes will make dependence bigger. So even though a correlation between genome size and dependence seems likely, as the authors confirm for vertically transmitted symbionts, such a correlation does not provide an answer to the question addressed.

Despite this problem, the whole text is written as if a phylogenetic comparative analysis will enable the discovery of causal factors for differences in dependence of hosts on symbionts across different symbioses. At best, such an analysis will provide correlations between dependence and certain factors. I find it misleading to present those correlations as causal relationships. As an example, transmission mode and symbiont function have the strongest correlation with symbiont dependence and this finding is described (in the abstract) as: “We found that both transmission mode and symbiont function were important in explaining (my italics) the evolution of host dependency, with the reductions in host fitness being greatest when nutrient-provisioning, vertically transmitted symbionts were removed.” I think this interpretation does not do justice to alternative causal explanations. For example, it seems likely that certain nutritional dependences have evolved after vertical transmission evolved, securing long-term interaction, and facilitating loss of redundancy in functions between both partners. In fact, flipping cause and consequence could make perfect sense (“We found that host dependency was important in explaining the evolution of both transmission mode and symbiont function, with the reductions in host fitness being greatest when nutrient-provisioning, vertically transmitted symbionts were removed.”)

We agree with Reviewer #1 that it is difficult to separate causes and consequences. We therefore have clarified the strengths and the limitations of our comparative approach, and highlight specific approaches needed to elucidate causation (lines 250-254). In our revised version, we have been strict in not implying causation.

It should be pointed out that vertical transmission must be uniparental in order to be vertical (and not just “from parent to offspring”), otherwise there is a horizontal component.

This is an important point as there are rare documented cases of paternal transmission of symbionts (Moran & Dunbar 2006, Damiani *et al* 2008, Zielinski *et al* 2009). In our database there are two examples of this phenomenon. Moran & Dunbar (2006) report the potential transfer of *Regiella insecticola* through the paternal line, leading to rare events of horizontal transmission. We also have identified a single example of biparental transmission in *Verminephrobacter*.

As our definition focuses on the dominant mode of symbiont transmission (lines 282-289) not the rare events, these cases would still be classified as vertical transmission. However to

confirm these examples of biparental/paternal transmission are not impacting our results, we ran our main analysis again with these two symbiont species assigned as ‘horizontal transmission’. Our main result, that vertical transmission is correlated with higher host dependence remained the same (Table S12 & S13). To clarify this point, we have addressed these cases in the main Methods section and the Supplementary methods.

Within the category of beneficial interaction, the authors focus on dependence. Why host-symbiont dependence? The interesting trait would be ‘beneficial’ or ‘detrimental’. But this is hard to measure.

Yes, Reviewer #1 is absolutely correct. The issue is that ‘benefit’ is potentially intertwined with ‘dependency’ in obligate symbioses (see Bennett & Moran 2015 *PNAS*). To clarify this issue, we have highlighted examples where benefit and dependency are not necessarily linked (lines 44-47).

The whole story is based on a phylogenetic comparative analysis of 106 host-bacterial symbioses. This sounds like a lot, but the strength of such analysis depends on the quality of the data. I think the authors need to point out the uncertainty with the phylogenetic estimate, and how robust their analysis is. Furthermore, how well has dependence been studied for the different examples included in this study.

We agree that the strength of our analysis depends on the quality of the data, as is true with any comparative study. We are confident about the robustness of our results for a number of reasons:

(i) We focus solely on experimental studies in order to gain as accurate estimates of the fitness effects of symbiosis as possible. This is especially important given the disparate taxa in our dataset.

(ii) Our analyses include a diverse range of host species. Therefore, it was important to account for their phylogenetic relationships. With respect to the symbiont phylogeny, this was created from 16S data to produce a single maximum likelihood tree. For the host phylogeny, we utilized the most up to date phylogenies from the Open Tree of Life (ToL) project. Phylogenies from ToL are created by synthesizing current phylogenetic evidence into a single phylogenetic tree for a given set of taxa. This approach does not produce a distribution of trees such as with Bayesian phylogenetic analysis that produce a posterior sample of trees. This means it is not possible to examine the influence of phylogenetic uncertainty on the results. However, this is unlikely to really influence our results as the species in our analysis are from very different and distinct phylogenetic groups and so error in estimating the topology of our phylogenetic tree expected to be minimal.

(iii) We have added new analyses to investigate the robustness of our conclusions to the ways fitness was measured across experimental studies. We found no significant differences between the methods used (Table S8 & S9). We have also included analyses on a smaller dataset where we restricted our data to fecundity fitness measurements as this is the most reliable measure of host fitness, rather than other measures such as size, survival or weight (Table S10 & S11). Our main conclusions remain the same even when we limit the dataset in this way. See section ‘Robustness of our results’ lines (198-213).

(iv) We have a two paragraph section (“Capturing the diversity of symbioses”) discussing possible limitations of our data set in the Discussion (lines 215-242).

Dependence or dependency? Both terms are used in the manuscript.

We have now edited the manuscript to be consistent.

Reviewer #2 (Remarks to the Author):

The manuscript explores evolutionary correlates of dependency in bacterial symbioses, a

topic of broad significance in biology. The authors present several very interesting findings, including links between transmission mode and the evolution of dependency and genome size reduction. Overall, I found this study to be very compelling, nicely executed, well written, and very big picture. I think that it will be a highly cited contribution. However, that being said, I have several major concerns:

In many cases, the number of datapoints used in the analyses is much larger than the number of species included. It was difficult to understand whether symbiont-host pair, species, or 'datapoint' was used as the unit of replication in the models. This made me worry about pseudoreplication and power inflation, as well as skewing results to be more heavily weighted by species that are represented by more papers in the literature. For example, for the analysis with genome size as the response variable and transmission mode as the predictor, 13 horizontally transmitted and 54 vertically transmitted symbioses were included, but 125 and 253 datapoints were used in the analysis, respectively (Figure 5 figure legend). It appears that the species means were not used in the analyses (n=66, as presented in the figure), or estimated as part of the model, but that all of the datapoints were used in the analyses (n=378) resulting in multiple datapoints per species being treated as independent. Tables S2 and S3 suggest the latter. This feels to me like an over inflation of power. Do results of the paper (in particular, the analyses presented in tables S2 and S3) hold if species are treated as the unit of replication (meaning there would be a maximum of 61 symbiont datapoints and 92 host datapoints in analyses)? Perhaps I am missing something here (and I apologize if this is the case), but this feels to me like a major issue to be resolved.

We agree with reviewer #2 that more clarity is needed in explaining our approach. We have edited the manuscript to include a better explanation of the statistical methods used to account for non-independence and repeated measures. Specifically, in our analyses, we include three random effects: (1) symbiont/host phylogeny (to account for non-independence due to shared ancestry), (2) the study which we got fitness measures from (to account for multiple measurements per study), and (3) the symbiont or host species (to account for repeated measures).

We can confirm that multiple data points per species are not treated as independent, as we include symbiont and host species as random effects to account for this. In the statistical results tables, we have now clarified sample sizes by including the number of actual data points (fitness measures) and the number of symbioses (host-symbiont pairs) in each figure legend.

As the authors mention, the cost/benefit that a host accrues from a given symbiosis changes across different environments. As such, we would expect the 'change in host fitness with symbiont removed' to be a function of the context in which each study was conducted. In particular, it is important to know whether removal experiments occurred in the presence or absence of stressors (i.e., pathogens or parasites in the case of defensive mutualisms, or nutrient stress in the case of nutritional mutualisms). To be fair, the authors touch on this issue nicely at the end of the manuscript for defensive symbionts (but strangely, not nutritional symbionts, stating earlier that the benefit is ubiquitous as a rule, which I disagree with, e.g., nitrogen fixers in high nitrogen environments). However, the authors do not report how their dataset is likely affected by from this issue. What number of studies in this dataset reported the presence of stressors? Is it possible to run analyses on a subset of data using only studies conducted in the face of stressors? Does this change the results?

Roughly 10% of the datasets collected contain information on stressors. Unfortunately this is not enough coverage to run a full analysis. We have now added a line to the discussion highlighting that the effect of stressors should be tested more frequently.

The methods state (lines 292-294): “We considered parameter estimates statistically significant when 95% credible intervals did not include 0 and pMCMC values (number of simulated cases that are > 0 or < 0 corrected for finite number of MCMC samples) were less than 0.05.” However, in several cases, reports are treated as significant when 95% CI appears to overlap 0:

- (1) For the difference in genome size –dependency relationship slope between vertically and horizontally transmitted symbionts (figure 4A, line 153), the confidence interval overlaps 0.*
- (2) Again line 111 for the difference in fitness reduction of removing symbionts in vertically vs horizontally transmitted symbionts.*
- (3) Again line 184*

We are grateful to referee for highlighting this issue and we have ensured that we do not use the term significant where credible intervals overlap 0. As stated in the methods, we use both the credible intervals and the pMCMC values to infer whether our results are statistically significant.

Minor Comments:

Love figures 1 & 2! Very nice looking and clear.

Brief information on how the literature search was conducted is needed in the main methods. Line 262 suggests that the authors used Papers 2 (“we searched the literature using Papers 2 (as described above)”, but this information is only in the supplemental methods.

We agree this requires more clarity and have included a summary of how the literature search was conducted in the main methods, rather than in the supplementary information.

*In figure 4 there is one vertically transmitted symbiont that is an outlier for genome size (I think it is *Rhodococcus rhodnii*, based on the supplementary table). However, this datapoint appears to be missing from figure 5. Why?*

We thank reviewer #2 for highlighting this error. We have edited the figure to include the missing datapoint.

Line 121 of supplement states “(see supplementary R code for details)” (also see line 135) but no R code is included. Including the code would really help the readers understand exactly what was done, as well as the replicate results.

We have included the R code in the Supplementary information.

If the specific and jargony term ‘evolutionary rabbit holes’ is used in the abstract, it should probably be discussed (or at least defined) in the main text as well.

We have removed ‘evolutionary rabbit holes’ from the abstract, and replaced with less jargony language.

Reviewer #3 (Remarks to the Author):

This is an interesting paper. Its strength comes from its breadth. I am not aware of any other work which tries to broadly tie so many important factors (transmission mode, symbiosis age, intra- vs. extra-cellularity, etc.) together with such a broad sampling of taxa across the tree of life. The methods all seem reasonable to me, and the conclusions seem solid. Some of the findings are not very surprising, but the authors are correct in saying that while many of these ideas have been out there for a long time, they have never been carefully tested. So there is utility here, and I think this paper will be cited because of that.

I have little to criticize about the paper, aside from two issues. The first relates to the analysis of genome size and host dependence. There is some discussion of the age of the symbiosis being important (and potentially confounding). This is mostly on lines 163-170. The authors conclude that “there was no association between age of symbiosis and host dependency.” I would be very interested to know if this conclusion seems to be affected by symbiont replacement. The authors have not discussed symbiont replacement in the paper, but it is known for some of the symbioses under study, so it might be worth checking if that affects these conclusions.

We agree with reviewer #3 that this is an interesting question. To our knowledge, there are 6 species that have potentially undergone symbiont replacement: 4 symbiont species for which we have host fitness data (*Nardonella*, *Coxiella*, *Riesia*, *Wolbachia*, *Pantoea*) and another 2 for which we have genome data (*Zinderia* and *Hodgkinia*). However, based on these 6 species, we only have 1 phylogenetic contrast between vertically and horizontally transmitted symbionts, this would not be enough to draw any conclusions. Furthermore, by their nature, facultative symbionts will swap hosts occasionally, so replacement cannot be determined in these cases.

The second relates to the abundance of insect symbioses in the study. I know the authors corrected for phylogeny in some cases, but should the large number of insect systems be cause for concern here? This might be appropriate, since most animals are in fact insects, but I wonder if randomly down-sampling insect examples might change the results.

We agree, and now discuss (line 222-229), that there is bias towards insects in the analysis. However, for our comparative analysis, the important point is not the number of species per se, but rather the phylogenetic contrasts. Comparative analyses are useful in that they give weights to where there are, for example, differences in transmission mode, rather than just number of species. The insects contain only 2 contrasts in transmission mode – *Frankliniella* and *Riptortus* are the only insect hosts that transmit their symbionts horizontally. Consequently, the phylogenetic analyses are controlling for, not giving undue weight to, where we have multiple closely related species where transmission mode does not vary.

We now stress that this is the first quantitative analysis to include a broad a range of organisms from “insects, plants, fungi, molluscs, arachnids and worms” (line 90). While our analyses may change as more data from different species (and therefore more contrasts emerge), these results hold for the current data set, which is as much any comparative study can contribute.

REVIEWERS' COMMENTS:

Reviewer #1 (Remarks to the Author):

The authors have addressed most of my concerns satisfactorily, but not completely. My main criticism focused on the limitations of a comparative analysis to determine causation, as the authors seemed to imply. In their response, the authors say that they agree with that criticism and that they have adapted the manuscript and 'have been strict in not implying causation'. Indeed, the authors have done this at most places, for example in the abstract (line 24-25) "We found that both transmission mode and symbiont function were correlated with host dependence, with reductions in host fitness being greatest when nutrient-provisioning, vertically transmitted symbionts were removed."

However, the authors are not consequent, for example, in the abstract they also write (line 22-24):

"We carried out phylogenetic comparative analyses on 106 host-bacterial symbioses to test the effect of symbiont function, transmission mode and genome size on host dependence." To my opinion, the way this is written still is suggestive of causation, so this sentence should be changed to: "We carried out phylogenetic comparative analyses on 106 host-bacterial symbioses to test for correlations between on the one hand symbiont function, transmission mode and genome size and, on the other hand, host dependence."

Similarly, in line 127, the authors write: "We then tested the effect of symbiont function within each transmission mode." Again, to me this suggests causation, which is more than just correlation.

I think this issue can be resolved by rewording those sentences.

Reviewer #2 (Remarks to the Author):

The revised manuscript addressed my methodological concerns as best as it could have. However, the authors were unable to deal with my criticism surrounding the context dependency in their measure of dependence. Because dependency was measured as the reduction in fitness when a symbiont was experimentally removed, the conditions in which the experimental reduction in host fitness was measured would matter greatly. The authors communicated that they were unable to deal with this issue because of a lack of sufficient data on the conditions of experiments. Unfortunately, this drastically reduces the ability to interpret the 'dependence' measure used the study (the major response variable in the analyses). I don't think that this is a total fatal flaw of the study, but it is a rather large caveat. The authors discuss this caveat at the end of the paper in relation to defense symbioses, but not for nutritional interactions. The authors should discuss the issue up front (when they first describe their measure of dependency). They should explicitly say that the dependency that the authors are examining is likely plastic across environmental conditions, so studying its evolution at this broad scale is inherently problematic. However, there may be no obvious reason why this plasticity should be biased with respect to transition mode,

etc. So perhaps the manuscript could (and should?) argue that this variation would come out in the wash in this broad analysis.

Additionally, re-reading the manuscript and the reviews, I am not convinced that the authors adequately dealt with the (very astute) criticisms of reviewer one. Primarily, the issue of causation vs correlation was not sufficiently dealt with in the new draft. I agree with the first reviewer that this is a major problem with the manuscript. In the 'Response to Reviewers' the authors say that they take out language concerning causation, but I did not feel that they adequately removed the language in the new version.

This manuscript touches on interesting questions, but I did not find the revisions particularly compelling given these two issues.

Reviewer #3 (Remarks to the Author):

The authors have addressed my concerns.

Point-by-point responses to reviewers comments

Reviewer #1 (Remarks to the Author):

The authors have addressed most of my concerns satisfactorily, but not completely. My main criticism focused on the limitations of a comparative analysis to determine causation, as the authors seemed to imply. In their response, the authors say that they agree with that criticism and that they have adapted the manuscript and 'have been strict in not implying causation'. Indeed, the authors have done this at most places, for example in the abstract (line 24-25) "We found that both transmission mode and symbiont function were correlated with host dependence, with reductions in host fitness being greatest when nutrient-provisioning, vertically transmitted symbionts were removed."

However, the authors are not consequent, for example, in the abstract they also write (line 22-24): "We carried out phylogenetic comparative analyses on 106 host-bacterial symbioses to test the effect of symbiont function, transmission mode and genome size on host dependence." To my opinion, the way this is written still is suggestive of causation, so this sentence should be changed to: "We carried out phylogenetic comparative analyses on 106 host-bacterial symbioses to test for correlations between on the one hand symbiont function, transmission mode and genome size and, on the other hand, host dependence."

We have reworded this sentence so that we only suggest correlation, and do not imply causation: "Here, we perform phylogenetic comparative analyses on 106 unique host-bacterial symbioses to investigate the relationships between symbiont function, transmission mode, genome size and host dependence" (lines 26-28).

Similarly, in line 127, the authors write: "We then tested the effect of symbiont function within each transmission mode." Again, to me this suggests causation, which is more than just correlation.

We have edited this sentence so that we no longer imply causation: "We then tested whether symbiont function was correlated with host dependence within each transmission mode" (lines 144-145).

I think this issue can be resolved by rewording those sentences.

Reviewer #2 (Remarks to the Author):

The revised manuscript addressed my methodological concerns as best as it could have. However, the authors were unable to deal with my criticism surrounding the context dependency in their measure of dependence. Because dependency was measured as the reduction in fitness when a symbiont was experimentally removed, the conditions in which the experimental reduction in host fitness was measured would matter greatly. The authors communicated that they were unable to deal with this issue because of a lack of sufficient data on the conditions of experiments. Unfortunately, this drastically reduces the ability to interpret the 'dependence' measure used the study (the major response variable in the analyses). I don't think that this is a total fatal flaw of the study, but it is a rather large caveat. The authors discuss this caveat at the end of the paper in relation to defense symbioses, but not for nutritional interactions. The authors should discuss the issue up front (when they first describe their measure of dependency).

We have raised this issue up front at the first instance in lines 99-102. We have included a more detailed discussion further on in the manuscript, because this is best done after analysing the data (see lines 250-264).

They should explicitly say that the dependency that the authors are examining is likely plastic across environmental conditions, so studying its evolution at this broad scale is inherently problematic. However, there may be no obvious reason why this plasticity should be biased with respect to transition mode, etc. So perhaps the manuscript could (and should?) argue that this variation would come out in the wash in this broad analysis.

We have included sections explicitly addressing this issue (lines 250-264, and lines 266-270).

Additionally, re-reading the manuscript and the reviews, I am not convinced that the authors adequately dealt with the (very astute) criticisms of reviewer one. Primarily, the issue of causation vs correlation was not sufficiently dealt with in the new draft. I agree with the first reviewer that this is a major problem with the manuscript. In the 'Response to Reviewers' the authors say that they take out language concerning causation, but I did not feel that they adequately removed the language in the new version.

We have further dealt with comments about causation, in line with Reviewer 1.

This manuscript touches on interesting questions, but I did not find the revisions particularly compelling given these two issues.

Reviewer #3 (Remarks to the Author):

The authors have addressed my concerns.